# Radiation-Induced EMT of Adipose-Derived Stem Cells in 3D Organotypic Culture via Notch Signaling Pathway

**DOI:** 10.3390/biology14091306

**Published:** 2025-09-22

**Authors:** Seon Jeong Choi, Meesun Kim, Kyung Tae Chung, Tae Gen Son

**Affiliations:** 1Research Center, Dongnam Institute of Radiological and Medical Science, 40 Jwadong-gil, Jangan-eup, Gijang-gun, Busan 46033, Republic of Korea; tjswjd5954@dirams.re.kr (S.J.C.); mskim@dirams.re.kr (M.K.); 2Department of Clinical Laboratory Science, Dong-Eui University, 176 Eomgwangno, Busanjin-gu, Busan 47340, Republic of Korea; kchung@deu.ac.kr

**Keywords:** 3D cell culture, irradiation, epithelial–mesenchymal transition, organoid, adipose-derived stem cell, Notch signaling pathway

## Abstract

Radiation is an essential tool in medicine, especially in cancer therapy, but it can also trigger unwanted changes in normal cells. One important change is called the “epithelial-to-mesenchymal transition,” a natural process in which cells lose their stable structure and become more mobile. While this process helps in normal development and wound healing, it can lead to tissue scarring and the spread of cancer when it continues for too long. In our study, we explored how radiation influences this transition in the short term. To achieve this, we used a three-dimensional cell model that more closely mimics the natural environment of tissues compared to traditional flat cultures. This approach allowed us to observe radiation-induced changes in cell behavior and organization in a setting that better reflects real biological conditions. Our findings highlight that radiation can rapidly trigger cellular transitions that may contribute to long-term complications. By improving our understanding of these early events, this research may support the development of cancer treatments that are both more effective and safer for healthy tissues.

## 1. Introduction

Two-dimensional (2D) cell culture has long served as a cornerstone for in vitro studies due to its simplicity and reproducibility. However, 2D systems insufficiently recapitulate the structural and biochemical complexity of living tissues, such as extracellular matrix (ECM) composition, mechanical cues, and cell–cell interactions [1,2]. This gap has driven the development of three-dimensional (3D) culture systems, which better mimic in vivo conditions and allow for more physiologically relevant investigations [3].

Among these platforms, organotypic cultures and organoids—self-organizing structures derived from stem or progenitor cells—provide powerful models for studying tissue organization, differentiation, and disease processes [3,4]. Adipose-derived stem cells (ASCs) represent an attractive cell source for organoid generation because they are abundant, easily accessible, and exhibit robust proliferation and multipotency. Importantly, while ASCs in 2D culture are prone to phenotypic drift and senescence [5], 3D conditions can preserve stemness and even induce epithelial-like features, highlighting the role of dimensionality in regulating cellular identity [6,7].

A key aspect of cellular plasticity involves the dynamic balance between mesenchymal–epithelial transition (MET) and epithelial–mesenchymal transition (EMT). These processes are tightly regulated by signaling pathways such as Notch, which is critically involved in development, cancer progression, and fibrosis [8,9]. Ionizing radiation—one of the most widely used modalities in cancer therapy—has been reported to promote EMT by inducing genotoxic stress and altering the tumor microenvironment. However, most previous studies have focused on cancer cell lines or conventional 2D cultures [10,11,12], and little is known about how radiation influences EMT and Notch signaling in ASC-derived 3D organoid systems.

Beyond ionizing radiation, several reports have shown that non-ionizing radiation can also trigger mitochondrial and endoplasmic reticulum stress, thereby influencing cellular plasticity and stem cell differentiation [13,14]. These observations suggest that radiation, regardless of energy type, may act on subcellular compartments to modulate EMT/MET dynamics. However, whether such mechanisms operate in ASC-derived 3D organoids has not been explored.

Addressing this knowledge gap is important because ASC-derived organoids not only provide a preclinical model to study stem cell plasticity but also represent a promising platform to investigate radiation-induced tissue responses and identify potential therapeutic targets. In this study, we employed a 3D organotypic culture model of murine ASCs previously shown to display MET-like features [8]. We investigated whether ionizing radiation induces EMT in this context and whether the Notch pathway mediates this transition. By doing so, we provide new insights into radiation-driven cellular plasticity and establish ASC organoids as tractable models for preclinical radiobiology research.

## 2. Materials and Methods

### 2.1. ASC-to-Epithelial-like Transition in 3D Culture

ASCs were isolated from male BALB/c mice (Central Laboratory Animals, Seoul, Republic of Korea) following CO_2_ euthanasia. Inguinal and gonadal white adipose tissues were excised under aseptic conditions, rinsed with HBSS (Gibco, Thermo Fisher Scientific, Waltham, MA, USA) supplemented with 2% Antibiotic-Antimycotic Solution (Gibco, Thermo Fisher Scientific, Waltham, MA, USA), and minced using sterile surgical blades (Feather, Osaka, Japan).

The tissue fragments were digested in 2 mg/mL collagenase type I (Gibco, Thermo Fisher Scientific, Waltham, MA, USA) at 37 °C for 30 min with shaking at 150 rpm. The resulting suspension was filtered through a 100 μm cell strainer (SPL Life Sciences, Pocheon, Republic of Korea) and centrifuged at 1000 rpm for 5 min at 4 °C. The supernatant containing adipocytes was carefully removed using a wide-bore pipette (Corning Inc., Corning, NY, USA), and the remaining stromal vascular fraction (SVF) pellet was resuspended in DMEM/F12 medium supplemented with 10% fetal bovine serum (Gibco, Thermo Fisher Scientific, Waltham, MA, USA) and 1% Antibiotic-Antimycotic Solution. The SVF cells were resuspended in Matrigel (Corning Inc., Corning, NY, USA), and approximately (5 × 10^3^ cells/well) were seeded in 12-well tissue culture plates (Corning Inc., Corning, NY, USA) in dome-shaped droplets. The culture medium was composed of Advanced DMEM/F-12 without FBS, supplemented with 50 ng/mL of recombinant murine epidermal growth factor (mEGF), 100 ng/mL of recombinant human Noggin (hNoggin), and 100 ng/mL of recombinant human fibroblast growth factor-7 (FGF-7) (all from PeproTech, Cranbury, NJ, USA), along with 3 µM of CHIR99021, 10 mM of nicotinamide, 1 mM of N-acetyl-L-cysteine (NAC), and 10 µM of Y-27632 dihydrochloride (all from Sigma-Aldrich, Merck KGaA, Darmstadt, Germany). Cultures were incubated at 37 °C with 5% CO_2_, and the medium was changed every 5 days. Organoids (ASC-to-epithelial-like cells) between passages 9 and 11 were used for experiments.

### 2.2. Determination of Irradiation Dose

To identify an effective gamma irradiation dose, two-dimensional ASCs isolated from BALB/c mice were seeded at (5 × 10^3^ cells/well) in 24-well plates and exposed to 0, 1, 2, 4, or 8 Gy using a Biobeam 8000 irradiator (Gamma-Service Medical GmbH, Leipzig, Germany). Cell viability was evaluated at 48, 72 and 96 h post-irradiation using the CellTiter-Glo^®^ assay (Promega). The dose that resulted in the greatest reduction in viability was applied to three-dimensional ASC organoids, which were thawed, dissociated into single cells, and seeded at (2.5 × 10^4^ cells/well) in Matrigel domes.

The organoids were then irradiated at 0 or 8 Gy, and viability was reassessed at 96 h using the same assay.

### 2.3. Cytotoxicity-Based GSI Concentration Selection

To determine an appropriate concentration of γ-secretase inhibitor (GSI) for use in organoid experiments, cytotoxicity was assessed in 3D ASC organoids. Cells were seeded at (2.5 × 10^4^ cells/well) in 24-well plates and treated with GSI at a final concentration of 0, 10, 20, or 40 µM. After 5 days of incubation, 250 µL of phosphate-buffered saline (PBS) and 250 µL of CellTiter-Glo^®^ Luminescent Cell Viability Assay reagent (Promega, Madison, WI, USA) were added to each well. The mixture was incubated for 10 min in the dark, and luminescence was measured using a microplate reader (Tristar3, BERTHOLD Technologies, Bad Wildbad, Germany). The luminescence intensity was used as an indicator of cell viability, and the concentration showing the least cytotoxic effect was selected for subsequent studies.

### 2.4. Quantitative Reverse Transcription Polymerase Chain Reaction

Total RNA was extracted from 3D adipose-derived stem cell (ASC) organoids using the RNeasy Mini Kit (Qiagen, Hilden, Germany) according to the manufacturer’s protocol. Complementary DNA (cDNA) was synthesized from the isolated RNA using the PrimeScript™ RT Master Mix (Takara Bio Inc., Shiga, Japan). Each qPCR reaction was prepared in a total volume of 20 µL, consisting of 3 µL of cDNA, 10 µL of SYBR Green qPCR master mix (TOPreal™ SYBR Green qPCR UDG PreMIX_Manual; Enzynomics, Daejeon, Republic of Korea), 2 µL of primer mix (forward and reverse), and 5 µL of distilled water. Amplification was performed using the LightCycler^®^ 96 Real-Time PCR System (Roche Diagnostics, Basel, Switzerland) under the following cycling conditions: an initial denaturation at 95 °C for 30 s, followed by 45 cycles of 95 °C for 30 s, 55 °C for 30 s, and 72 °C for 30 s. Gene expression was calculated using the comparative 2^−ΔΔCt^ method, with normalization to a housekeeping gene. The primers used for RT-qPCR are listed in Appendix A.

The primers used for qRT-PCR were as follows: *E-cadherin* forward, 5′-GCT CTC ATC ATC GCC ACA G-3′, and reverse, 5′-GAT ATG AGG CTG TGG GTT CC-3′; *Fibronectin* forward, 5′-GAG ACT TCT CTC CTC AAT GGT G-3′, and reverse, 5′-CCT ATT GAT CCC AGA CCA AAC C-3′; *Twist1* forward, 5′-GTG GAC AGA GAT TCC CAG AG-3′, and reverse, 5′-CTT CGT CAA AAA GTG GGG TGG-3′; *Notch1* forward, 5′-GTG CTC TGA TGG ACG ACA AT-3′, and reverse, 5′-GTC TGA TCA CTC AGG TCA GG-3′; *Jagged1* forward, 5′-CTT GGG TCT GTT GCT TGG TG-3′, and reverse, 5′-TGG CTC CGT GTT TCT CGA TG-3′; *Fra-1* forward, 5′-GCT GCA GAA GCA GAA GGA AC-3′, and reverse, 5′-GTA CGG GTC CTG GAG AAA G-3′.

### 2.5. Western Blot Analysis

Cells were lysed using PRO-PREP™ buffer (iNtRON Biotechnology, Seongnam, Republic of Korea) with protease inhibitors and centrifuged at 13,000 rpm for 5 min at 4 °C. Protein concentration was determined using a bicinchoninic acid (BCA) assay kit (Thermo Fisher Scientific, Waltham, MA, USA). Equal amounts of protein were denatured at 100 °C for 5 min and separated using SDS-PAGE on 4–12% Bis-Tris Plus gels (Invitrogen, Carlsbad, CA, USA), followed by transfer to polyvinylidene difluoride (PVDF) membranes (GE Healthcare, Chicago, IL, USA). Membranes were blocked with 5% skim milk in TBST and incubated overnight at 4 °C with primary antibodies against E-cadherin, vimentin, and fibronectin (Cell Signaling Technology, Danvers, MA, USA), each diluted at a ratio of 1:1000. After washing, membranes were incubated with horseradish peroxidase (HRP)-conjugated secondary antibodies. Protein bands were visualized using an enhanced chemiluminescence (ECL) detection reagent (Cytiva, Buckinghamshire, UK) and imaged with the ImageQuant™ 800 system (Cytiva, Marlborough, MA, USA). GAPDH antibody (rabbit monoclonal; Cell Signaling Technology) was used as a loading control.

### 2.6. Statistical Analyses

All experiments were independently repeated at least three times using separate cell cultures. Data were analyzed using GraphPad Prism version 8.4.3 (GraphPad Software, San Diego, CA, USA). Depending on the experimental design, statistical significance was assessed using either a one-way or two-way analysis of variance (ANOVA). A *p*-value less than 0.05 was considered statistically significant.

## 3. Results

### 3.1. Morphological Distinctions Between 2D and 3D ASC Cultures

ASCs cultured under 2D conditions displayed typical fibroblast-like morphology, forming a uniform monolayer with sparse cell–cell contact (Figure 1A). In contrast, 3D organotypic cultures in Matrigel produced compact, spheroid aggregates that matured into denser and more stabilized structures over time (Figure 1B).

### 3.2. Dimensionality-Dependent Expression of Epithelial, Mesenchymal, and Notch Markers

To assess the impact of culture dimensionality on ASC phenotype, the expression levels of epithelial, mesenchymal, and Notch-related genes were evaluated using RT-qPCR and Western blotting. Mesenchymal markers such as α-SMA, fibronectin, N-cadherin, and vimentin, along with EMT transcription factors (Slug, Twist1, Twist2, and Snail), were more highly expressed in 2D cultures. Among these, α-SMA, fibronectin, and Snail showed statistically significant elevation (Figure 2A). Conversely, epithelial markers including E-cadherin, CK18, and EpCAM were more abundantly expressed in 3D organoids, with CK18 and E-cadherin confirmed at the protein level (Figure 2C). The expression of Notch1, Notch2, Jagged1, DLL1, and downstream effector Fra-1 was also significantly higher in 3D cultures, suggesting baseline Notch activation in the 3D context (Figure 2B).

### 3.3. Radiation-Induced EMT and Notch Activation in 3D ASC Organoids

A dose–response analysis of γ-irradiated 2D ASCs revealed a time-dependent reduction in cell viability, with 8 Gy causing the most pronounced effect at 96 h post-irradiation (49.50 ± 6.50%, *p* < 0.0001) (Figure 3A). Applying this dose to 3D ASC organoids yielded a moderately higher survival rate (61.02 ± 5.77%) (Figure 3B). Irradiated 3D organoids exhibited significantly increased expression of fibronectin and Twist1, indicating EMT induction, while CK18 was modestly elevated (Figure 4A). Western blotting confirmed similar trends (Figure 4C). Notch signaling analysis revealed upregulation of Notch1, Notch3, Jagged1, and Fra-1, suggesting pathway activation in response to irradiation (Figure 4B), and levels of STAT3 and NF-κB remained unchanged.

### 3.4. Inhibition of Notch Signaling Attenuates Radiation-Induced EMT

To evaluate the functional role of Notch signaling, optimal dosing of the γ-secretase inhibitor (GSI) was determined. While 10 μM and 20 μM GSI maintained high cell viability (>90%), 40 μM significantly impaired viability (Figure 5A,B). Thus, 20 μM GSI was selected for subsequent experiments. Pretreatment with GSI prior to irradiation led to a significant reduction in fibronectin and Twist1 expression compared to irradiated controls, without affecting epithelial marker levels (Figure 6A). Western blot analysis showed corresponding protein-level trends (Figure 6B). Additionally, GSI suppressed the radiation-induced expression of Notch1, Jagged1, and Fra-1, supporting the involvement of Notch signaling in radiation-mediated EMT (Figure 6A).

## 4. Discussion

Ionizing radiation not only induces DNA damage but also activates diverse intracellular signaling pathways, including those that promote epithelial–mesenchymal transition (EMT), a key process in cancer progression, fibrosis, and tissue remodeling. EMT is orchestrated by several core signaling cascades, notably the TGF-β, Wnt, Hedgehog, and Notch pathways [10]. Among these, the Notch signaling pathway—mediated by direct cell–cell contact—is evolutionarily conserved and governs cell fate, survival, and stemness. Its aberrant activation has been reported in various cancers and fibrotic conditions [11,12]. In this study, we employed a physiologically relevant 3D ASC organoid model to explore radiation-induced EMT and the role of Notch signaling. Unlike conventional 2D cultures or tumor-derived lines, this model better mimics in vivo microenvironments and overcomes limitations of traditional culture systems [13].

Comparative analyses revealed that 3D-cultured ASCs retained epithelial characteristics, including higher E-cadherin and CK18 expression, while 2D ASCs exhibited increased mesenchymal traits (e.g., α-SMA, and fibronectin) and upregulated EMT transcription factors (Slug, Twist1, and Snail), with Snail showing the most significant increase. These findings align with previous studies reporting that 3D cultures preserve mesenchymal stem cell characteristics and enhance cell–ECM interactions [8]. Radiation response also differed significantly by culture format. Three-dimensional ASC organoids exhibited higher resistance to 8 Gy γ-radiation than 2D counterparts (61.0% vs. 49.5% viability at 96 h), suggesting structural or ECM-mediated protection—consistent with earlier findings in 3D systems [13]. Based on these results, 8 Gy was chosen as the optimal dose for subsequent mechanistic assays.

Beyond transcriptional regulators, EMT is also influenced by functional changes in subcellular compartments. Ionizing radiation has been shown to perturb mitochondrial dynamics, increase reactive oxygen species (ROS) production, and induce endoplasmic reticulum (ER) stress responses, which collectively contribute to cytoskeletal remodeling and EMT progression. Such organelle-level stress can act synergistically with Notch signaling to reinforce mesenchymal traits. Interestingly, similar mechanisms have been reported with non-ionizing radiation, where changes in mitochondrial bioenergetics, redox balance, and ER homeostasis alter stem cell plasticity and differentiation [14]. These observations suggest that both ionizing and non-ionizing radiation converge on organelle stress pathways to regulate EMT, although the upstream triggers differ—DNA damage and genotoxic stress are observed in ionizing radiation, while bioenergetic or mechanotransduction-related cues are observed in non-ionizing radiation [14].

Radiation exposure induced EMT in 3D ASC organoids, as indicated by the upregulation of fibronectin and Twist1. Concurrently, Notch1, Notch3, Jagged1, and the downstream effector Fra-1, were significantly increased, implicating Notch pathway activation in EMT progression. Specifically, Notch1 has been implicated in self-renewal, while Notch3 contributes to differentiation [15], suggesting a coordinated stress response. To confirm this, we pharmacologically inhibited Notch activation using γ-secretase inhibitor (GSI) prior to irradiation. GSI pretreatment significantly suppressed fibronectin, Twist1, Notch1, Jagged1, and Fra-1 expression, indicating that EMT induction is at least partially dependent on Notch signaling. Fra-1, a member of the AP-1 complex, has been identified as a critical downstream target of Notch in EMT and tumor invasion [16]. Importantly, GSI at 10–20 μM demonstrated minimal cytotoxicity and preserved the 3D organoid structure over five days, supporting its utility for long-term inhibition studies in this model.

While our study primarily focused on short-term radiation responses, it is also important to consider the long-term functional implications as sustained EMT has been implicated in promoting tissue fibrosis, chronic remodeling, and increased metastatic potential in vivo [17,18]. PersistentIn alterations in Notch signaling may disrupt the balance of stem cell plasticity, leading to aberrant differentiation or loss of regenerative capacity over time [19,20,21]. These long-term effects highlight the importance of investigating radiation-induced EMT not only as a transient phenomenon but also as a driver of chronic tissue dysfunction. Importantly, ASC-derived organoids offer a tractable platform to model such prolonged responses under controlled conditions and to evaluate potential therapeutic strategies aimed at mitigating radiation-induced remodeling and stem cell exhaustion.

Taken together, our results support the notion that the Notch1–Jagged1–Fra-1 axis is a key driver of radiation-induced EMT in ASCs. Radiation promotes EMT through Notch activation, and GSI effectively abrogates this transition—highlighting Notch inhibition as a promising therapeutic strategy against radiation-induced fibrosis and tissue remodeling. This study also reinforces the value of the ASC-based 3D organoid system as a robust platform for investigating radiation biology and EMT-associated diseases. Unlike cancer-derived organoids, ASC organoids better replicate normal tissue dynamics making them suitable for preclinical modeling and drug testing. While our study provides novel insights into radiation-induced EMT and Notch signaling using mouse-derived ASCs, it is important to recognize that interspecies differences in gene regulation, signaling dynamics, and microenvironmental responses may limit the direct translation of these findings to human biology. Nevertheless, the 3D organotypic culture system employed here recapitulates essential aspects of stem cell plasticity, supporting its value as a mechanistic platform. Future studies using human-derived ASCs or patient-derived organoids will be essential to validate the relevance of these observations in a clinical context.

## 5. Limitations

This study’s limitations include being conducted in vitro using murine-derived ASCs, and extrapolation to human biology requiring validation in human organoid systems. Additionally, while the Notch pathway was the primary focus, cross-talk with other EMT-related pathways such as TGF-β, Wnt, and Hedgehog was not explored [10]. Future studies should assess interactions among these pathways in human-derived models to enhance translational relevance.

## 6. Conclusions

This study demonstrates that γ-radiation promotes epithelial–mesenchymal transition (EMT) in 3D adipose-derived stem cell (ASC) organoids through pathological activation of the Notch signaling pathway. Pharmacological inhibition using a γ-secretase inhibitor (GSI) effectively suppressed radiation-induced EMT, highlighting Notch as a potential therapeutic target for mitigating radiation-associated tissue remodeling. Moreover, the 3D ASC organoid model offers a physiologically relevant platform for studying radiation-induced fibrosis and signaling dynamics. Future studies incorporating in vivo validation, longitudinal analysis, and pathway cross-talk will be essential to extend these findings toward translational applications.

## Figures and Tables

**Figure 1 biology-14-01306-f001:**
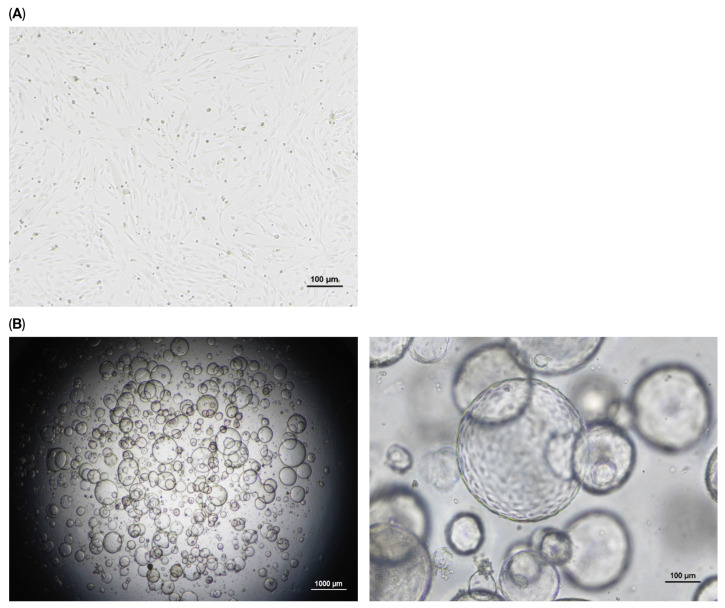
Morphological comparison of ASCs under 2D and 3D culture conditions. (**A**) ASCs cultured in 2D conditions exhibited fibroblast-like morphology, spreading radially across the plate. Images were captured on day 3 at ×4 magnification. Scale bar = 100 μm. (**B**) In 3D culture with Matrigel embedding, ASCs formed compact, organoid-like spherical aggregates. Images were acquired on day 7 at ×1 and ×4 magnification. Scale bars: (**B. left**) 1000 μm, and (**B. right**) 100 μm.

**Figure 2 biology-14-01306-f002:**
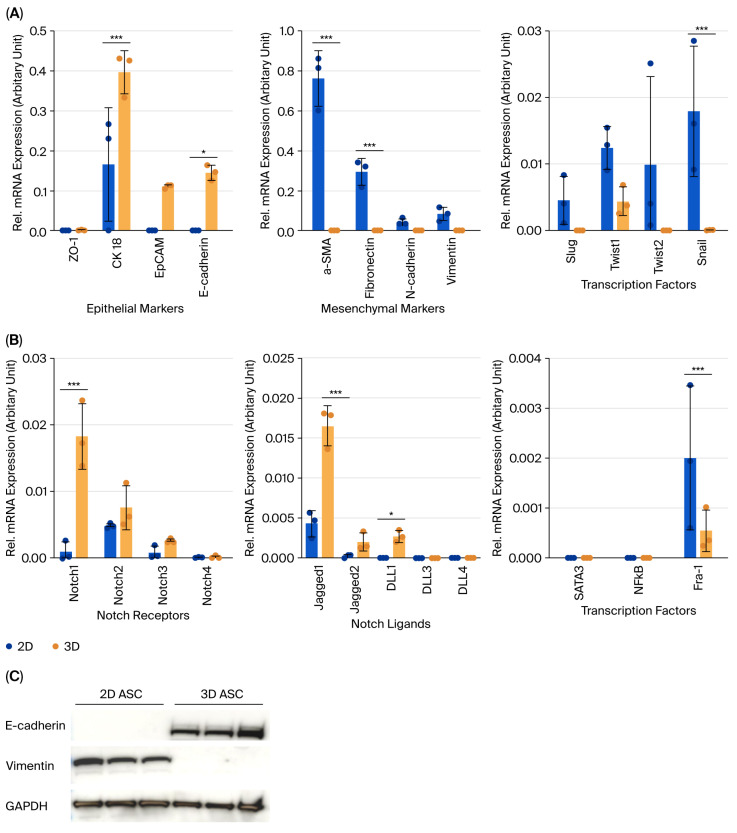
EMT and Notch-related gene expression profiles in 2D ASCs and 3D ASC organoids. (**A**) RT-qPCR analysis of epithelial (CK18, E-cadherin) and mesenchymal (α-SMA, fibronectin) markers. Epithelial markers were upregulated in 3D organoids, while mesenchymal markers were more prominent in 2D ASCs. (**B**) Notch pathway components (Notch1, Jagged1, and Fra-1) showed higher expression in 3D organoids compared to 2D cultures. (**C**) Western blot analysis revealed increased E-cadherin (120 kDa) in 3D organoids and elevated vimentin (57 kDa) in 2D ASCs. GAPDH (37 kDa) was used as a loading control. Data are shown as mean ± SD (*n* = 3). Statistical analysis: two-way ANOVA. * *p* < 0.05, *** *p* < 0.001.

**Figure 3 biology-14-01306-f003:**
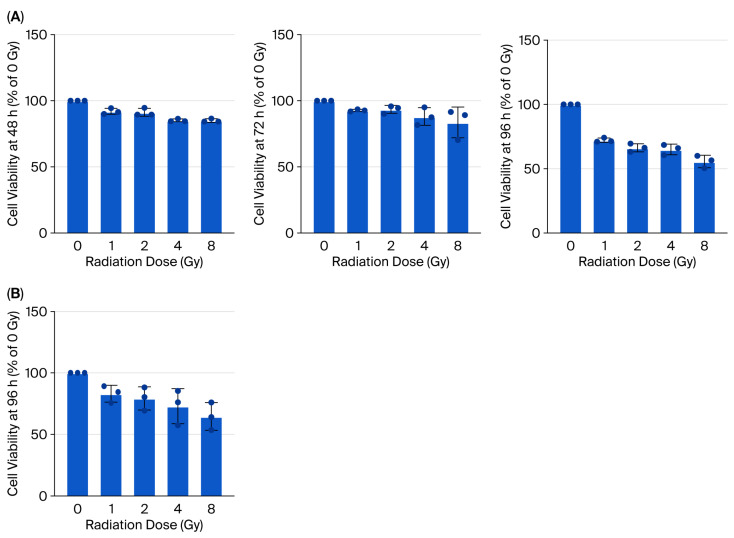
Cell viability of 2D and 3D ASCs following γ-irradiation. (**A**) Viability of 2D ASCs at 48, 72, and 96 h after exposure to 0, 1, 2, 4, and 8 Gy of γ-irradiation. (**B**) Viability of 3D ASC organoids 96 h after exposure to 0, 1, 2, 4, 8 Gy. Data are presented as mean ± SD (*n* = 3). Statistical analysis: one-way ANOVA.

**Figure 4 biology-14-01306-f004:**
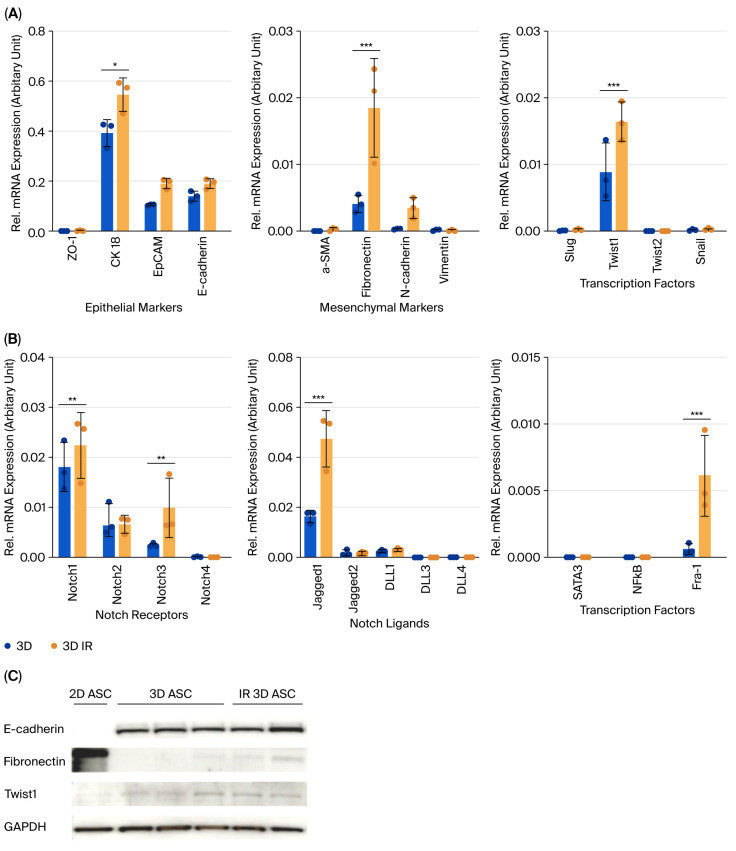
Changes in EMT and Notch marker expression in 3D ASC organoids following γ-irradiation. (**A**) RT-qPCR analysis of epithelial and mesenchymal markers. CK18, fibronectin, and N-cadherin were upregulated post-irradiation. EMT transcription factors (Slug, Twist1/2, and Snail) also showed increased expression, particularly Twist1. (**B**) Notch signaling genes (Notch1, Notch3, Jagged1, and Fra-1) were significantly upregulated following irradiation. (**C**) Western blot analysis of E-cadherin, fibronectin, and Twist1 across 2D ASCs, 3D organoids, and irradiated 3D organoids. GAPDH served as the loading control. Data are shown as mean ± SD (*n* = 3). Two-way ANOVA was used for statistical comparisons. * *p* < 0.05, ** *p* < 0.01, and *** *p* < 0.001.

**Figure 5 biology-14-01306-f005:**
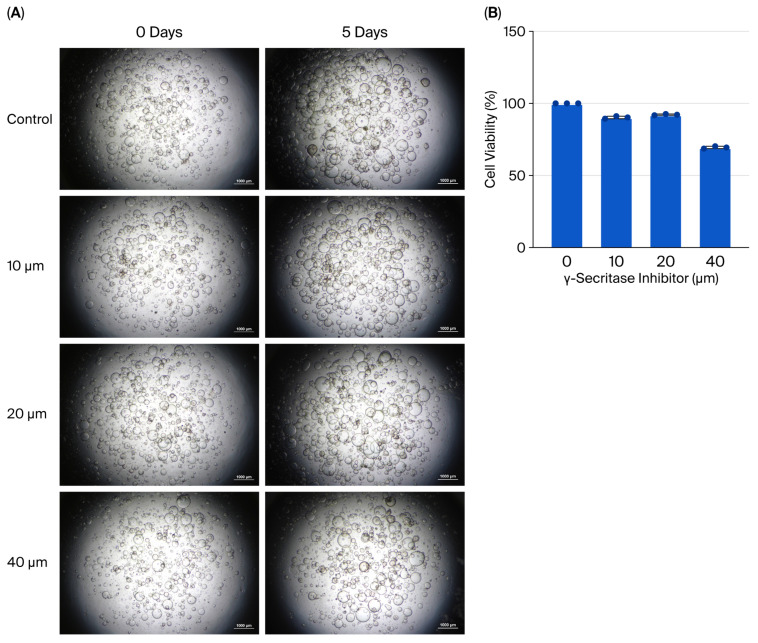
Effect of γ-secretase inhibitor (GSI) on the morphology and viability of 3D ASC organoids. (**A**) Bright-field images of 3D ASC organoids before and after 5-day treatment with 0 (control), 10, 20, and 40 μM GSI. Images were taken on days 0 and 5. Morphological changes were visually evaluated. (**B**) Cell viability was assessed after 5 days of GSI treatment. A significant reduction in viability was observed at 40 μM. Data are shown as mean ± SD (*n* = 3).

**Figure 6 biology-14-01306-f006:**
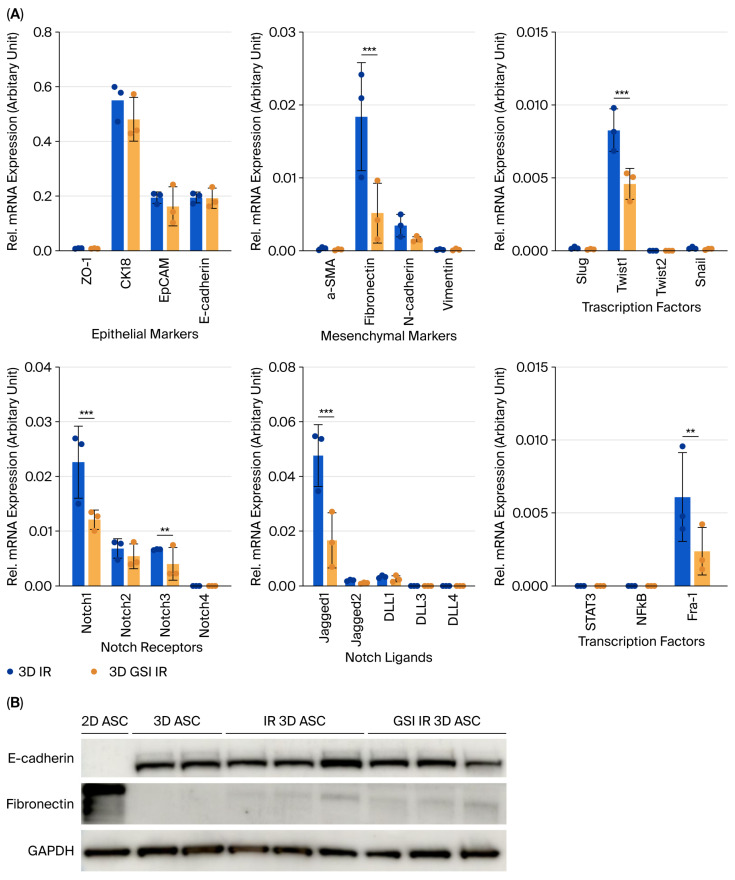
Effect of GSI pretreatment on EMT and Notch signaling in irradiated 3D ASC organoids. (**A**) EMT-related gene expression was broadly downregulated in GSI-pretreated irradiated organoids compared to the irradiated-only group. Both epithelial (ZO-1, CK18, EpCAM, and E-cadherin) and mesenchymal (α-SMA, fibronectin, N-cadherin, and vimentin) markers were decreased. EMT-associated transcription factors (Slug, Twist1/2, and Snail) were also downregulated. Notch signaling components (Notch1, Jagged1, and Fra-1) were significantly suppressed by GSI pretreatment. (**B**) Western blotting showed no significant differences in E-cadherin and fibronectin protein levels among 2D ASCs, 3D organoids, irradiated 3D organoids, and GSI-pretreated irradiated organoids. GAPDH was used as a loading control. Data are shown as mean ± SD (*n* = 3). Two-way ANOVA was used for statistical comparisons. ** *p* < 0.01, and *** *p* < 0.001.

## Data Availability

The data supporting the findings of this study are available from the corresponding author upon reasonable request.

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
