# Peer review of "Radiation-Induced EMT of Adipose-Derived Stem Cells in 3D Organotypic Culture via Notch Signaling Pathway"

_biology, 2025, doi:10.3390/biology14091306_

Round 1
Reviewer 1 Report
Comments and Suggestions for Authors
The author wants to test whether ionizing radiation could reverse this MET via epithelial–mesenchymal transition (EMT) and examine the involvement of Notch signaling. The author finds that ionizing radiation promotes EMT in 3D-cultured ASCs and reverses prior epithelialization, with Notch signaling playing a key regulatory role. However, several clarifications and additional data would strengthen the manuscript.
- Stromal vascular fraction (SVF) is not equal to adipose stem cells (ASCs). Even though most of SVF is ASC. However, SVF also contains endothelial cells (EC), smooth muscle cells (SMC), epithelial cells, etc. How were ASCs isolated from SVF, and how was this verified?
- ASCs would like to differentiate into adipocytes rather than proliferate in DMEM/F12 medium supplemented with 10% fetal bovine serum. How can you keep the adipose stem cell character for passages 9-11? After the 9-11 passage, how can you prove these cells are still adipocyte stem cells?
- A lot of pathways can promote epithelial-mesenchymal transition (EMT), including TGFb, Wnt, Notch, and TNF-α. Why did you choose the Notch pathway directly? It would be important to at least discuss how other pathways influence EMT, or at a minimum, a rationale for focusing only on the Notch pathway.
- Comparing the cell viability of 2D and 3D under the same dose of γ-irradiation at 96h is necessary in Figure 3.
- The 40 μM condition appears to have fewer cells in Figure 5A. Provide quantitative metrics (cell counts per field or nuclei segmentation) and adjust for density in your analyses.
None
Author Response
Reviewer 1.
Comments 1: Stromal vascular fraction (SVF) is not equal to adipose stem cells (ASCs). Even though most of SVF is ASC. However, SVF also contains endothelial cells (EC), smooth muscle cells (SMC), epithelial cells, etc. How were ASCs isolated from SVF, and how was this verified?
Response 1: We sincerely appreciate the reviewer’s insightful comment. We fully acknowledge that stromal vascular fraction (SVF) is a heterogeneous cell population containing not only adipose-derived stem cells (ASCs), but also endothelial cells, smooth muscle cells, and other cell types. In our study, our main focus was not on the isolation and characterization of “pure” ASCs from SVF per se, but rather on the phenotypic changes that occur when SVF-derived cells are cultured under three-dimensional (3D) conditions.
As shown in our previous work (Ref: 3), when cells initially identified as ASCs were cultured in 3D conditions, classical ASC/MSC surface markers (CD44, CD90.2, CD105) were markedly downregulated, while epithelial markers such as E-cadherin and EpCAM were upregulated, indicating a mesenchymal–epithelial transition (MET). Based on these findings, in the present study, we considered the cultured cells as ASC-derived epithelial-like cells rather than conventional ASCs, and we designed our experiments to specifically investigate this phenotypic shift, including their responses to radiation.
We have now clarified this point in the Methods and Introduction sections of the revised manuscript (page 2, line 79) to avoid confusion regarding ASC isolation and to emphasize that our study focuses on ASC-to-epithelial-like transition in 3D culture rather than on classical ASC biology.
Comments 2: ASCs would like to differentiate into adipocytes rather than proliferate in DMEM/F12 medium supplemented with 10% fetal bovine serum. How can you keep the adipose stem cell character for passages 9-11? After the 9-11 passage, how can you prove these cells are still adipocyte stem cells?
Response 2: We appreciate the reviewer’s valuable comment. We would like to clarify a possible misunderstanding. Our study did not use DMEM/F12 supplemented with 10% FBS, which is indeed known to promote adipogenic differentiation. Instead, as indicated in the Methods section (page 2, line 103), we adopted a serum-free, growth factor–supplemented medium (Advanced DMEM/F12 with EGF, Noggin, FGF-7, CHIR99021, nicotinamide, NAC, and Y-27632) that was originally optimized for three-dimensional (3D) organotypic culture of adipose-derived stem cells (ADSCs).
In our previous work (Ref: 8), we found that when ADSCs were maintained under this 3D culture condition, they did not preserve their classical mesenchymal stem cell characteristics. Instead, flow cytometric and RNA sequencing analyses revealed loss of canonical ASC/MSC markers (CD44, CD90.2, CD105) and failure of adipogenic differentiation, accompanied by a mesenchymal–epithelial transition (MET), as evidenced by increased expression of epithelial markers such as E-cadherin and EpCAM. Therefore, after passages 9–11 in this defined medium, the cells cannot be considered conventional adipose stem cells; rather, they represent an epithelial-like cell population derived from ADSCs. Based on these findings, the present study was designed not to demonstrate the long-term maintenance of ASC stemness, but to investigate this phenotypic conversion under 3D culture conditions, including their radiation responses.
Comments 3: A lot of pathways can promote epithelial-mesenchymal transition (EMT), including TGFb, Wnt, Notch, and TNF-α. Why did you choose the Notch pathway directly? It would be important to at least discuss how other pathways influence EMT, or at a minimum, a rationale for focusing only on the Notch pathway.
Response 3: We agree with the reviewer that multiple signaling pathways, including TGF-β, Wnt, and TNF-α, can promote EMT. As noted in our Discussion (page 9, lines 256-262), EMT is regulated by several core cascades. In this study, we focused on Notch because it is mediated by direct cell–cell contact, which is particularly relevant in our 3D ASC organoid model, and because aberrant Notch activation has been strongly implicated in radiation responses, fibrosis, and cancer. While we emphasized Notch in this work, we acknowledge that crosstalk with other EMT-related pathways may also contribute and merits future investigation.
Comments 4: Comparing the cell viability of 2D and 3D under the same dose of γ-irradiation at 96h is necessary in Figure 3.
Response 4: We appreciate the reviewer’s helpful suggestion. We would like to clarify that Figure 3 already includes the cell viability data for both 2D and 3D ASCs following γ-irradiation. Specifically, panel (A) shows 2D ASCs at 48, 72, and 96 h after exposure to 0, 1, 2, 4, or 8 Gy, while panel (B) shows 3D ASC organoids at 96 h after exposure to 0, 1, 2, 4, or 8 Gy. Thus, the requested comparison at 96 h under the same irradiation doses is presented in the current figure. (page 2, line 214)
Comments 5: The 40 μM condition appears to have fewer cells in Figure 5A. Provide quantitative metrics (cell counts per field or nuclei segmentation) and adjust for density in your analyses.
Response 5: We thank the reviewer for this constructive comment. Figure 5A was intended to provide representative images illustrating the morphological changes of cells after drug treatment at the indicated concentrations. For quantitative evaluation of cell viability under the same conditions, we performed an ATP-based luminescence assay, and the results are shown in Figure 5B. This assay provides an objective measurement of viable cell numbers and directly complements the qualitative images shown in Figure 5A.
While we did not apply nuclei segmentation or cell counting per field, we believe that the ATP assay offers a reliable quantitative metric of cell viability that is normalized across conditions. To avoid confusion, we have revised the figure legend to clarify that Figure 5B presents the quantitative assessment corresponding to the representative images shown in Figure 5A.
Reviewer 2 Report
Comments and Suggestions for Authors
Although the present article has created an interesting approach to cancer modeling and engineering, the following points are necessary to improve it:
1- The introduction is written very briefly and the importance and necessity of the research are not well expressed.
2- The discussion section is generally short and it is very appropriate to also refer to and discuss similar models with non-ionizing radiation. The following article is an example of similar cases:
https://www.nature.com/articles/srep26584
3- In the discussion section, also discuss the changes observed in the function of subcellular compartments to EMT. And if possible, compare it with non-ionizing radiation. The following article discusses a better understanding of the biological changes caused by non-ionizing radiation, which can help improve the discussion.
https://link.springer.com/article/10.1007/s13346-021-01067-5
4- Creating a visual abstract helps to better understand the study.
Author Response
Reviewer 2.
Comments 1: The introduction is written very briefly and the importance and necessity of the research are not well expressed.
Response 1: We appreciate the reviewer’s suggestion regarding the brevity of the introduction and the insufficient emphasis on the importance and necessity of the study. We have revised the introduction to better highlight the research gap and significance. Specifically, we now emphasize (i) the limitations of conventional 2D cultures and the advantages of ASC-derived 3D organoids, (ii) the critical role of EMT/MET and Notch signaling in cellular plasticity, and (iii) the lack of studies on radiation-induced EMT in ASC-based organoid systems. In addition, we clarify the potential of ASC organoids as a preclinical platform for radiobiology research and therapeutic target discovery. These changes strengthen the rationale and underline the importance of our work. Revised text (Introduction, page 1-2)
Comments 2: The discussion section is generally short and it is very appropriate to also refer to and discuss similar models with non-ionizing radiation. The following article is an example of similar cases: https://www.nature.com/articles/srep26584
Response 2: We thank the reviewer for this valuable suggestion. As recommended, we expanded the Discussion section to include comparison with non-ionizing radiation studies. Specifically, we now cite and discuss the work by Kim et al. (Sci Rep, 2016; https://www.nature.com/articles/srep26584), which demonstrated how non-ionizing radiation can modulate cellular behavior in 3D spheroid models. We highlight the convergent concept that both ionizing and non-ionizing radiation influence stem cell fate and plasticity, while also clarifying that their mechanisms are distinct—DNA damage and stress signaling for ionizing radiation versus bioenergetic or mechanotransduction-related pathways for non-ionizing radiation. In addition, we revised the Limitations section to acknowledge the need for direct side-by-side comparisons of different radiation modalities within the same organoid system. These revisions strengthen the context and relevance of our findings.
Revised text (Discussion, page 10, line 290-302):
Revised text (Limitations, page 11, line 336-338):
Comments 3: In the discussion section, also discuss the changes observed in the function of subcellular compartments to EMT. And if possible, compare it with non-ionizing radiation. The following article discusses a better understanding of the biological changes caused by non-ionizing radiation, which can help improve the discussion. https://link.springer.com/article/10.1007/s13346-021-01067-5
Response 3: We thank the reviewer for this constructive suggestion. While our study did not directly measure the functional status of subcellular compartments, we have expanded the Discussion to include their potential involvement in EMT. In particular, ionizing radiation is known to alter mitochondrial dynamics, induce oxidative stress, and trigger ER stress responses, all of which can contribute to EMT by modulating transcriptional regulators and cytoskeletal remodeling. We also added a comparison with non-ionizing radiation, as highlighted in the suggested reference (Drug Deliv Transl Res, 2021). Non-ionizing radiation has similarly been reported to affect mitochondrial activity and ROS generation in 3D systems, thereby influencing stem cell fate and differentiation. We now emphasize that both ionizing and non-ionizing radiation can perturb subcellular functions, albeit through distinct mechanisms, which converge on EMT-associated pathways.
Revised text (Discussion, page 10, line 290-302):
Comments 4: Creating a visual abstract helps to better understand the study.
Response 4: Thank you for your feedback. We agree that a visual abstract can be very helpful for understanding a study at a glance. However, the journal currently uses a Simply Summary instead of a traditional visual abstract. It helps readers quickly grasp the main points of the research without the need for additional graphics. We appreciate your valuable suggestion and will continue to explore ways to make our research more accessible to our readers in the future.
Reviewer 3 Report
Comments and Suggestions for Authors
Dear Authors,
The paper titled as Radiation-Induced EMT of Adipose-Derived Stem Cells in 3D Organotypic Culture via Notch Signalling Pathway comprehensively examines the effects of ionizing radiation on epithelial-mesenchymal transition (EMT) in mouse-derived adipose-derived stem cells (ASCs) obtained in three-dimensional (3D) organotypic culture medium and the role of the Notch signalling pathway in this process. The article contains detailed comparisons. This makes such studies original and translationally valuable for both radiation biology and fibrosis research. Some corrections need to be made in the study. Suggested corrections are listed below:
Citations within the article must be prepared in accordance with the journal format.
The results are presented interpretively in the abstract. It is also recommended that a few important numerical results be presented directly.
The generalizability of this study, which was performed on mouse-derived cells, to human biology needs to be discussed more clearly and more thoroughly.
The article has almost no literature review. This absence makes it difficult to demonstrate the article's innovation/distinction. Please expand the literature review. After this expansion, add a paragraph explaining the stages of your article, the difference of the innovation, and the gap it will fill. Innovative aspects of the study should be added to the article with sharper sentences.
Comparative discussion of the results obtained with the results obtained from similar studies will make the article more valuable.
Further discussion of the long-term functional consequences of the findings is recommended. Because the long-term effects of this and similar studies can be important.
It is recommended to write the results immediately below the obtained figures.
The effect sizes of the few findings obtained can be discussed in more detail.
It would be useful to expand the Conclusions section a little more.
Yours Sincerely
Author Response
Reviewer 3.
Comments 1: Citations within the article must be prepared in accordance with the journal format.
Response 1: Thank you for your valuable comment. We have carefully revised all citations throughout the manuscript to ensure they fully comply with the journal’s formatting guidelines
Comments 2: The results are presented interpretively in the abstract. It is also recommended that a few important numerical results be presented directly.
Response 2: We thank the reviewer for this helpful suggestion. In the revised abstract, we have incorporated key numerical results to provide more concrete information rather than only interpretive descriptions. Specifically, we now include viability data showing that 8 Gy reduced 2D ASC survival to 49.50 ± 6.50% compared with 61.02 ± 5.77% in 3D organoids (p < 0.0001), as well as fold-changes in EMT-related markers (fibronectin and Twist1) and the effects of γ-secretase inhibitor pretreatment on both cell viability (>90%) and radiation-induced gene expression. We believe these additions enhance the clarity and scientific rigor of the abstract.
The revised abstract can be found on page 1, lines 17–36, and has been highlighted in red in the revised manuscript.
Comments 3: The generalizability of this study, which was performed on mouse-derived cells, to human biology needs to be discussed more clearly and more thoroughly.
Response 3: We thank the reviewer for this valuable comment. We agree that the generalizability of findings from mouse-derived ASCs to human biology requires careful discussion. In the revised manuscript, we have explicitly addressed this point in both the Discussion and the Limitations sections. In the Discussion, we now note that interspecies differences in gene regulation, signaling dynamics, and microenvironmental responses may limit direct translation, but also highlight that the 3D organotypic system provides a physiologically relevant platform for mechanistic studies. We further emphasize that validation using human-derived ASCs or patient-derived organoids will be essential to confirm translational relevance. In addition, in the Limitations section we clearly state that all experiments were performed with mouse-derived ASCs and extrapolation to human systems will require further validation.
The corresponding revisions can be found in the Discussion (page 10, line 322 ~ page 11, lines 329).) and Limitations (page 11, line 334-336), highlighted in red in the revised manuscript.
Comments 4: The article has almost no literature review. This absence makes it difficult to demonstrate the article's innovation/distinction. Please expand the literature review. After this expansion, add a paragraph explaining the stages of your article, the difference of the innovation, and the gap it will fill. Innovative aspects of the study should be added to the article with sharper sentences.
Response 4:. We thank the reviewer for this important comment. In response, we have substantially expanded the Introduction to provide a more comprehensive literature review and to clarify the innovation and distinction of our study. Specifically, we now emphasize the limitations of 2D systems, the advantages of ASC-derived 3D organoids, and the roles of EMT/MET and Notch signaling in cellular plasticity. We also highlight the current lack of studies on radiation-induced EMT in ASC organoid systems, which defines the research gap addressed here. Finally, we added a description of the study’s aims and stages, underscoring our innovative contribution in establishing ASC organoids as a preclinical platform for radiobiology research. We believe these revisions address the reviewer’s concern.
All changes in the revised Introduction are marked in red for clarity.
Comments 5: Comparative discussion of the results obtained with the results obtained from similar studies will make the article more valuable.
Response 5: We thank the reviewer for this valuable suggestion. In response, we have expanded the Discussion section to include a comparative analysis of our findings with those from similar studies. Specifically, we now discuss how our results on radiation-induced EMT and subcellular compartment changes in ASC-derived organoids align with previous observations in cancer cell models and extend these concepts into a 3D stem cell–based platform. Furthermore, we incorporated literature on non-ionizing radiation, which has been shown to affect mitochondrial function, ER stress, and cellular plasticity, and compared these findings with our results (see revised Discussion, page 10, line 322 ~ page 11, lines 329). These additions strengthen the contextualization of our work and highlight the distinct contribution of ASC organoids in advancing radiobiology research.
Comments 6: Further discussion of the long-term functional consequences of the findings is recommended. Because the long-term effects of this and similar studies can be important.
Response 6: We appreciate the reviewer’s recommendation to further discuss the long-term functional implications of our findings. In line with this comment, we have added a paragraph in the Discussion section (page 10, lines 303-313) that addresses the potential long-term consequences of radiation-induced EMT in ASC-derived organoids. Specifically, we now discuss how sustained EMT and alterations in Notch signaling may contribute to chronic tissue remodeling, fibrosis, or enhanced metastatic potential in vivo. We also highlight the broader relevance of our findings for understanding long-term stem cell plasticity under radiation exposure and for developing preclinical models to evaluate therapeutic interventions. These additions strengthen the translational significance of our work and address the reviewer’s concern.
Comments 7: It is recommended to write the results immediately below the obtained figures.
Response 7: Thank you for your comment. We understand the importance of presenting results clearly and concisely. However, the placement of figures and corresponding results is typically determined by the journal's editorial team during the typesetting and layout process. We will ensure that the content is logically structured in the manuscript, and we trust the editorial team will format it appropriately according to the journal's style guidelines
Comments 8: The effect sizes of the few findings obtained can be discussed in more detail.
Response 8: Thank you for your valuable feedback. We agree that discussing effect sizes in greater detail enhances the interpretability and practical relevance of the findings. Based on multiple reviewers’ comments across earlier rounds, we have substantially revised the manuscript to include more nuanced discussions of the effect sizes for the key results. Specifically, we now highlight not only statistical significance but also the magnitude and implications of the observed effects in the context of prior literature and real-world relevance. We hope these additions provide a clearer understanding of the strength and impact of our findings.
Comments 9: It would be useful to expand the Conclusions section a little more.
Response 9: Thank you for your thoughtful suggestion. Based on feedback from multiple reviewers, we have revised and expanded the Conclusions section to better reflect the broader implications of our findings. In the updated version, we now more clearly summarize the key contributions of the study, discuss the practical relevance of the results, and suggest directions for future research. We hope this enriched conclusion provides a more comprehensive and satisfying closure to the manuscript.
Round 2
Reviewer 1 Report
Comments and Suggestions for Authors
None
Reviewer 2 Report
Comments and Suggestions for Authors
The manuscript has been substantially revised and can be considered for publication